# Exploring Caregivers’ Perspectives and Perceived Acceptability of a Mobile-Based Telemonitoring Program to Support Pregnant Women at High-Risk for Preeclampsia in Karachi, Pakistan: A Qualitative Descriptive Study

**DOI:** 10.3390/healthcare11030392

**Published:** 2023-01-30

**Authors:** Anam Shahil Feroz, Salima Nazim Meghani, Haleema Yasmin, Sarah Saleem, Zulfiqar Bhutta, Hajraa Arshad, Emily Seto

**Affiliations:** 1Community Health Sciences Department, Aga Khan University, Karachi 74800, Sindh, Pakistan; 2Institute of Health Policy Management and Evaluation, The University of Toronto, Toronto, ON M5T 3M6, Canada; 3School of Nursing & Midwifery, The Aga Khan University, Karachi 74800, Sindh, Pakistan; 4Department of Obstetrics and Gynecology, Jinnah Postgraduate Medical Center, Karachi 75510, Sindh, Pakistan; 5Centre for Global Child Health, SickKids, Toronto, ON M5G 1X8, Canada; 6Center of Excellence in Women and Child Health, Aga Khan University, Karachi 74800, Sindh, Pakistan; 7Dalla Lana School of Public Health, The University of Toronto, Toronto, ON M5T 3M7, Canada; 8Medical College, Aga Khan University, Karachi 74800, Sindh, Pakistan; 9Centre for Digital Therapeutics, University Health Network, Toronto, ON M5G 2C4, Canada

**Keywords:** caregivers, telemonitoring, preeclampsia, Pakistan, qualitative, pregnant women

## Abstract

Very little is known about the perspectives of the caregivers on the use of telemonitoring (TM) interventions in low-middle-income countries. Understanding caregivers’ perspectives on TM interventions is crucial, given that caregiving activities are correlated with the social, emotional, and clinical outcomes among pregnant women. This study aims to explore caregivers’ perspectives and perceived acceptability of a mobile phone-based TM program to support pregnant women at high-risk for preeclampsia. A qualitative description design was used to conduct and analyze 28 semi-structured interviews with a diverse group of caregivers. The study was conducted at the Jinnah Post Graduate Medical Center, Karachi, Pakistan. The caregivers were identified through purposive sampling and additional caregivers were interviewed until the point of data saturation. The conventional content analysis technique was used to analyze digital audio recordings of the caregiver interviews. All caregivers embraced the proposed mobile phone-based TM program because they perceived many benefits, including a reduction in caregivers’ anxiety and workload, increased convenience, and cost-effectiveness. However, the caregivers cited several caveats to the future implementation of the TM program including the inability of some women and caregivers to use the TM program and the poor acceptance of the TM system among less educated and non-tech savvy families. Our study recommends developing a TM program to reduce the caregiver stress and workload, designing a context-specific TM program using a user-centric approach, training caregivers on the use of the TM program, sensitizing caregivers on the benefits of the TM program, and developing a low-cost TM program to maximize access.

## 1. Introduction

In Pakistan, about one-third of the maternal mortality in the tertiary-level facilities is attributable to preeclampsia/eclampsia (PE/E) [1]. The high maternal mortality from PE/E results from a lack of early risk detection, diagnosis, and treatment of pregnant women at high-risk for PE [1]. Telemonitoring (TM) is a potential tool to support pregnant women at high-risk for PE; it refers to the use of information technology to routinely monitor patients at a distance [2]. TM for PE involves pregnant women monitoring their blood pressure at home with the help of devices (e.g., mobile phone apps and blood pressure machines) for real-time monitoring by providers [3]. Studies have recommended TM for high-risk pregnant women because of its benefits, such as better blood pressure control, early risk identification and treatment, fewer hospital visits, and cost savings [4,5,6,7].

Despite growing evidence that TM programs are beneficial for high-risk pregnant women [4,5,6,7], only a few pilot TM programs have been scaled-up and integrated into existing antenatal programs [3,5,7,8,9,10,11,12]. Thus, it is unclear whether the lack of scale-up of pilot TM programs reflects the limited efficacy of TM or the poor acceptability of TM among patients and caregivers. Most formative and summative digital health evaluations have focused on exploring the perceptions and experiences of pregnant women on the use of TM programs [12,13,14]. There is strong evidence that caregivers play an important and unique role to support pregnant women [15]; however, very little is known about the perspectives of the caregivers on the use of TM interventions. Understanding caregivers’ perspectives on TM interventions is crucial, given that caregivers are substantially involved in the care of pregnant women, and caregiving activities are found to be correlated with social, emotional, and clinical outcomes among pregnant women [15]. The roles and activities of caregivers are diverse and complex, and they likely depend on many factors associated with the patient, the caregiver, and the environment [15]. For instance, South-Asian caregivers, including husbands and in-laws, have a substantial influence on pregnancy-care decisions due to the patriarchal family system [16]. Our current study offers opportunities to engage caregivers of high-risk pregnant women to explore their views on the use of a mobile-based TM program.

This study will leverage foundational research completed by our team including a scoping review [17] and a needs assessment study [13]. Our review concluded that most TM interventions have been implemented in high-income countries, including the UK, USA, and Belgium [5,7,8,9,10,11,12], with a paucity of evidence on its use in low-middle-income countries (LMICs). Our needs assessment study with high-risk pregnant women and key stakeholders identified an opportunity for implementing a TM program to enable early risk detection and prompt treatment [13]. The needs assessment study also revealed several socio-cultural barriers that could potentially influence the use of the proposed TM program, including gendered imbalances in decision-making, restricted access to education and healthcare, and restricted women’s mobility. The interviewed high-risk pregnant women, nurses, clinicians, and digital health experts in the needs assessment study reported that pregnant women face restrictions, from in-laws and husbands, regarding the use of mobile phones and the internet. In addition, the study highlighted that women would require permission from husbands and in-laws before opting into the TM program, and family members might discourage the use of a TM program to support high-risk pregnant women [13]. Given the prominent role that informal caregivers play in the lives of pregnant women, we explored caregivers’ perspectives and the perceived acceptability of a mobile phone-based TM program. In this study, informal caregivers refer to family members, including male partners, mother-in-law, sister-in-law, and others in the family, who provide unpaid assistance related to pregnancy care and support, such as transportation and personal care. The insights gained from this study will help inform the development of the TM program as well as guide the future feasibility trial of the TM program in Karachi, Pakistan.

Research Question: What are the informal caregivers’ perspectives and perceived acceptability of a mobile phone-based TM to support pregnant women at high-risk for PE in Karachi, Pakistan?

## 2. Materials and Methods

### 2.1. Research Design and Setting Overview

A qualitative descriptive approach was used [18] to explore informal caregivers’ perspectives and perceived acceptability of a mobile phone-based TM program through semi-structured interviews with informal caregivers of pregnant women at high-risk for PE. The caregivers of pregnant women at high-risk for PE often include the husband, mother, mother-in-law, sister, and sister-in-law. This study was conducted at the Jinnah Post Graduate Medical Center (JPMC) [19], a 1650-bed tertiary-level public sector hospital, in Karachi, Pakistan, which provides hospital care to over one million patients yearly. The people visiting JPMC belong to low socioeconomic status and come from Karachi, Interior Sindh, Baluchistan, and other remote areas. This research focused on the outpatients of the JPMC obstetrics and gynecology (OB-GYN) department. The vast majority of high-risk pregnant women who attend the JPMC OB-GYN department are accompanied by their caregivers.

The study was approved by the JPMC Institutional Review Board (44379), and the University of Toronto Research Ethics Board (39948).

### 2.2. Proposed Mobile-Based Telemonitoring Program

During the interviews, the caregivers of high-risk pregnant women were provided with a theoretical description of a proposed mobile phone-based TM program. The proposed TM program includes the use of a Bluetooth-enabled home blood pressure device that is validated for use in pregnancy and a mobile app (in the Urdu language). The TM program will enable pregnant women to take their blood pressure every morning at home, and answer symptom questions using the mobile app. All enrolled women would receive real-time automated instructions based on their readings, such as taking additional blood pressure readings, calling the medical officer (i.e., the trained physician in Pakistan), or visiting the OB-GYN emergency department at JPMC. The real-time automated instructions would be delivered in Urdu text on the mobile app. The medical officer would receive alerts from the TM system if their patient’s blood pressure values were out of the target range. The medical officer would act as a central point person to communicate with the patients (phone calls or using the asynchronous app chat feature) and with the rest of the participant’s care team as needed (Figure 1).

### 2.3. Participant Recruitment and Eligibility Criteria

The qualitative description approach used a purposive sampling technique [20,21] to recruit caregivers for exploring differing perspectives on the use of a mobile phone-based TM program for supporting high-risk pregnant women. Due to COVID-19 restrictions, the caregivers were not allowed to accompany the pregnant women during the antenatal visit and were asked to wait outside the outpatient area of the JPMC OB-GYN department. The wait times typically ranged from one to four hours. The high-risk pregnant women were requested to help with the identification of the accompanied caregiver from the waiting area. The study coordinator then identified and recruited the eligible caregivers from the waiting area after explaining the study objectives and procedures to the study participants. Table 1 provides a list of the eligibility criteria for the caregivers of pregnant women at high-risk for PE.

### 2.4. Data Collection Methods

Interviews were conducted between February 2022 and March 2022. The research team and study participants had access to all necessary personal protective equipment to help prevent the risk of spreading the COVID-19 virus during the data collection. Eligible caregivers were informed about the study’s purpose and procedures (including the recording of interviews), and their willingness to participate in the study was ascertained by the study coordinator (SNM), a trained qualitative researcher. If the caregiver was unable to read the consent form, the study coordinator explained the consent form verbally in their local language. The caregivers who were unable to sign their names were asked to provide a thumbprint to mark their consent to participate.

The interviews were conducted by the study coordinator immediately after the caregivers provided consent. To avoid any disruption during the interview, and to ensure confidentiality, consenting caregivers were asked to move to a private room for the interview. We anticipated conducting and recording 20–25 interviews with informal caregivers of high-risk pregnant women; however, our study conducted 28 interviews with caregivers in order to achieve data saturation. Each interviewed caregiver corresponds to one pregnant woman.

A semi-structured interview guide was developed to interview the caregivers of pregnant women at high-risk for PE. The interview guide involved a general discussion about PE, causes of PE, caregivers’ perceptions towards the use of TM for supporting high-risk pregnant women, perceived acceptability of a mobile phone-based TM program, perceived benefits of TM for pregnant women and caregivers, and potential limitations or concerns related to TM for PE.

### 2.5. Data Analysis

Similar to Coffey et al.’s study [23], our study used digital audio recordings of the caregiver interviews for analysis purposes. The anonymized audio recordings were uploaded into NVivo Version 12 Plus (QSR International) to enable the easy and organized retrieval of data for analysis. The conventional content analysis approach [24] was used to inductively analyze all the interviews. The primary researcher (ASF) and second reviewer (SNM) independently coded all the interview recordings. The main themes and sub-themes were identified independently by the primary researcher and second reviewer and then discussed until agreement on the themes was achieved.

## 3. Results

A total of 28 semi-structured interviews were conducted to explore the caregivers’ perspectives and perceived acceptability of a mobile-based TM program to support pregnant women at high-risk for PE. Interviews were conducted with a diverse group of caregivers, with relationships including husband (n = 15), mother (n = 2), mother-in-law (n = 2), sister (n = 6), sister-in-law (n = 1), and wife of husband’s brother (n = 2). Each of these interviews lasted between 20 and 30 minutes. A total of 40 participants were approached by the study team, of which 28 caregivers agreed to participate. The demographic information for all the caregivers is illustrated in Table 2.

Based on the inductive analysis, four overarching themes were identified: caring for pregnant women at high-risk for PE; supporting caregivers through the use of the TM program; caregivers’ expectations of the TM program; and considerations for implementing the TM program. These themes and their sub-themes are summarized in Table 3.

### 3.1. Caring for Pregnant Women at High-Risk for Preeclampsia

#### 3.1.1. Caregivers’ Roles in Supporting High-Risk Pregnant Women

When asked about the role of caregivers in supporting high-risk pregnant women, almost all caregivers mentioned that they support women in a variety of ways, such as taking pregnant women to hospitals and clinics for regular antenatal care visits and blood pressure monitoring, taking women out for regular walks, preparing meals, taking care of children, supporting in household chores, purchasing prescribed medications, timely administering scheduled medicines, and ensuring a stress-free and comfortable environment at home. The caregivers reiterated that they do everything possible to keep them happy and stress-free, and routinely ask them about their concerns and anxieties that might be causing high blood pressure.


*I know how to care for her, and I try not to give her any stress. I take good care of her and provide her with everything she wants so she doesn’t get stressed. I also support her in taking care of our children so that she can stay stress-free. (Husband 14)*


The diet of pregnant women (less salt and more fruits and vegetables) was generally controlled by the females in the family. However, the males were responsible for shuttling women to the hospital for regular checkups and blood pressure monitoring. A few caregivers noted that they have blood pressure machines at home, and they monitor the blood pressure of high-risk pregnant women at least twice daily. Some caregivers mentioned that they have extended family to provide adequate support for high-risk pregnant women. The members of the extended family usually divide responsibilities amongst them to ensure adequate support for pregnant women.


*We live in a joint family, so I have support from my three brothers and their wives. Everyone takes good care of her and asks her about her health and wellbeing. (Husband 14)*


#### 3.1.2. Challenges in Caring for Pregnant Women at High-Risk for Preeclampsia

The interviews with caregivers revealed several challenges and difficulties that caregivers face while caring for high-risk pregnant women. Most caregivers mentioned that they encounter various challenges in making frequent hospital visits for continuous monitoring of high-risk pregnant women. Husbands noted that they experience disruption in their daily routine as they have to take off from work to be able to take women to the hospital for antenatal checkups and emergency visits. The husbands further articulated that reduced work hours influence their daily earnings, especially for daily-wage workers.


*The difficulty is to close the shop, rush home, take her to the hospital in time, purchase prescribed medicines, drop her home, and again go back to the work. Sometimes I feel helpless as I am alone and need to take care of her as well as run my shop to earn our living. (Husband 15)*


The majority of the caregivers reported difficulties in traveling to the hospital due to a poor public transportation system and a lack of access to private transport. The caregivers reported the average travel time is 45 to 90 min for reaching the JPMC, which they perceived as a long time and troublesome for high-risk pregnant women and themselves. The caregivers noted that traveling becomes more difficult and stressful when high-risk pregnant women are experiencing signs and symptoms of PE conditions, such as high blood pressure, headache, blurred vision, epigastric pain, and bleeding. The caregivers also expressed concern related to long wait times (5–7 h) in the hospital for seeing the doctor in an outpatient clinic.


*Traveling is a big issue especially when she is experiencing bleeding… in those circumstances, her husband rents a car and drives very slowly. Again, waiting at the hospital for seeing a doctor in a clinic is also a big challenge. (Brother’s Wife 27)*


The caregivers, mostly husbands, expressed their concern about leaving high-risk pregnant women alone at home in the daytime while they are at work. One female caregiver noted that she visits the pregnant woman regularly to care for her while her husband is away for work. A few male caregivers verbalized that they rely on God to watch over their wives while they were away.


*The difficulty is that… I am alone and do not have any support. When I leave for work, my wife is alone at home with my 4 years old son. God forbid if she becomes unconscious so there is no one to take care of her… my 4 years son would not be able to do anything for her. (Husband 17)*


In terms of blood pressure monitoring, a few caregivers mentioned that they have a blood pressure machine at home, but that they are not knowledgeable about operating them. Most caregivers, especially husbands, mentioned that they are only able to take women for blood pressure monitoring during the night hours at a nearby clinic after they return from work. They further mentioned that they have to pay a minimum PKR 100 every time they took a blood pressure reading at the clinic (often instructed by their care provider to take blood pressure readings twice daily), and often the clinic is closed by the time they get there.


*I am unable to take her for blood pressure monitoring in the nearby clinic for the last 3 to 4 days… since I get late from work. I leave at 0700am in the morning and return at 1100pm in the night and by that time all clinics are closed. (Husband 08)*


### 3.2. Supporting Caregivers through the Use of the Telemonitoring Program

#### 3.2.1. Reducing Caregiver Distress and Anxiety

Upon hearing about the proposed TM program, all caregivers stated that the TM program would aid in reducing the caregiver stress and anxiety through the regular monitoring of blood pressure and disease symptoms at home. Most of the husbands noted that they are under constant stress while they are at work because women are alone at home but with such a TM program, they would be able to work peacefully as women would be monitored by the hospital for their PE condition.


*I will not be stressed while I am at work as my wife will be monitored by the doctor for her health condition… I will not be worried even if she (his wife) is alone at home. (Husband 06)*


The husbands further mentioned that even if they get stuck at work, they would be at peace as women would be able to stay in touch with the doctor through the TM program. Other caregivers mentioned that the TM program would put their minds at ease given that caregivers would be able to track the daily blood pressures of the high-risk pregnant women.

#### 3.2.2. Reducing Caregiver Workload

All caregivers believed that the TM program could help in reducing caregivers’ workload to a large extent. The caregivers noted that the use of the TM program would aid in reducing the number of tasks, such as taking women to the clinic for blood pressure monitoring, recording daily blood pressure measurements, and making frequent hospital visits. Many caregivers, mostly husbands, verbalized that this TM program would save them a lot of time and energy as they would not have to take women to clinics for blood pressure monitoring after returning from their busy workday.


*It will be extremely helpful for me as my wife would be able to take care of herself and would be able to monitor her blood pressure and report it to the doctor through the app. This will save me a lot of time and effort as I will not need to take her to the clinic regularly for blood pressure monitoring… she will not need me during the daytime and would be able to reach out to the doctor by herself. (Sister 19)*


The caregivers noted that the pregnant women would no longer be dependent on their husbands, in-laws, mothers, and sisters for monitoring their condition. One husband mentioned that he would not have to run from his workplace to home in case women are experiencing high blood pressure and other symptoms. The husband articulated that the TM program would reduce a lot of his responsibilities as pregnant women would be able to self-monitor their blood pressure readings and report them to the doctor via the app.

#### 3.2.3. Convenient and Cost-Effective

Most caregivers felt that the TM program would be convenient because caregivers and pregnant women would not have to travel long distances and wait longer at the clinic to be able to see a doctor. Caregivers articulated that pregnant women would be able to easily monitor their blood pressures at home, at their own convenience, with the help of the TM program. Most importantly, caregivers, mostly mothers-in-law, believed that the TM program would allow caregivers to take care of the high-risk pregnant women, her kids, and other household chores while staying at home.


*We will not have to travel so far for seeing a doctor... the women will be monitored at home through the provided home blood pressure machine… we will be able to save a lot of time… we will not have to stand in long queues we will use the time to complete our household chores. (Mother-in-law 20)*


The caregivers further iterated that this would save them huge costs that they incur for traveling to the public hospital or clinic to receive pregnancy care. In addition, the caregivers noted that they would save the money they would otherwise incur for taking blood pressure readings at the nearby clinics (CAD 4 or PKR 600 daily for taking blood pressure readings twice daily).


*This machine will have many benefits. We will have a blood pressure monitoring facility at home… we will not have to pay 100Rs daily to the clinic for one-time blood pressure measurements. We will also save our transportation costs… and we will be able to use the savings for buying prescribed medicines and supplements. (Sister-in-law 24)*


### 3.3. Caregivers’ Expectations of the Telemonitoring Program

#### 3.3.1. Ease-of-Use of Telemonitoring Program

When asked about the expectations for a mobile phone-based TM program, most caregivers mentioned that the TM program should be easy-to-use for pregnant women and caregivers. All caregivers unanimously believed that the TM program should be very basic so that illiterate and non-tech savvy women can use it. One caregiver mentioned that the TM program should use a simple phone call function to allow non-tech-savvy women and caregivers to easily connect to the healthcare provider in an emergency. The caregivers further iterated that not all women have access to smartphones and the internet; thus, the hospital may need to provide access to technology for pregnant women.


*The machine should be very simple so that technologically illiterate people can use it… I know my wife would be able to use it, but it might be difficult for others who do not have the technology and are not literate (Husband 18).*


#### 3.3.2. Continuous Monitoring of Pregnant Women and Baby

The caregivers mentioned that they would like the TM program to continuously monitor pregnant women and babies, especially when the caregivers are away during the daytime. A few caregivers suggested that the TM program should also monitor high-risk pregnant women at night for high blood pressure and other symptoms to prevent complications and emergency department visits during the night. The caregivers suggested that the TM program should allow immediate contact with the healthcare provider in the case of an emergency. The caregivers further iterated that the TM program should include contact details of the healthcare providers who can be contacted for online advice and to provide medication prescriptions.


*The machine should include the contact number of the doctor so that the women can call the number and seek advice from the doctor on the medicines and home remedies for reducing blood pressure… the doctor should also be able to guide us about nearby hospitals for emergency visits. (Husband 17)*


### 3.4. Considerations for Implementing the Telemonitoring Program

#### 3.4.1. Caregivers and Community Acceptance of the Telemonitoring Program

All interviewed caregivers embraced the proposed TM program to support pregnant women at high-risk for PE. Most caregivers were very appreciative of the proposed TM program and mentioned that they would like to see such a TM program as soon as possible for supporting high-risk pregnant women. When asked about caregivers’ readiness to allow women to use TM, female caregivers mentioned that the women may require permission from their husbands for using such a TM program. The husbands noted that their wives would be allowed to use the TM program given that it would be beneficial for their high-risk pregnancy and the baby. However, a few husbands mentioned that they would seek advice from the elders in the family before approving the use of a TM program.


*I think we will have to seek permission from elders in our home like my mother, my father, and my grandmother… I believe they will be supportive of the use of this machine… since this machine is trying to help high-risk pregnant women. (Husband 06)*


Most caregivers noted that the TM program would be readily accepted by the majority of pregnant women, their caregivers, and the wider community because of the increasing use of mobile phones and the internet among women. A few caregivers explained the influence of sociocultural factors on the adoption of the TM program among families and communities. The caregivers unanimously believed that the educated and tech-savvy families would readily accept the use of the TM program, as opposed to the families and communities who were less educated and non-tech savvy. The caregivers mentioned that the less educated families might not trust the TM program and may anticipate harm from the use of such a TM program. The caregivers iterated that the resistance for the use of the TM program would only be for the initial few months because the families would begin to accept it once they recognize the value of the TM. The caregivers suggested that the hospital would need to conduct sensitization activities, such as awareness sessions and workshops, to convince conservative families and communities about the use of such a TM program.


*I will give permission to my daughter-in-law for the use of this machine but there might be some families that will not allow you to monitor pregnant women from home and you would need to give awareness to such families on the usefulness of this machine… and then they might be able to accept it. (Mother-in-law 20)*


#### 3.4.2. Training of Pregnant Women and Caregivers on Telemonitoring Program Use

When asked about the technological literacy to use the TM program, all caregivers unanimously believed that the young, educated, and tech-savvy pregnant women would be better able to learn and use the TM program as opposed to older women, who are less educated and non-tech savvy. Nonetheless, it was mentioned that such women would be able to receive help from immediate family members or caregivers to participate in a TM program.


*Some women are literate, and some are illiterate, so there would be some problems and you will have to guide them and show them how the machine can be used… I know the use of smartphones so I will be able to help my wife. (Husband 07)*


The caregivers, mostly husbands, mentioned that they would be able to train pregnant women on how to use the TM program. Some caregivers mentioned that the pregnant women would require training and guidance from the hospital on the use of the TM program. The caregivers further suggested training pregnant women as well as accompanying caregivers on the use of the TM program.

#### 3.4.3. Cost of Telemonitoring Program

When asked about the caregivers’ willingness to pay for the proposed TM program, most caregivers mentioned that they would not be able to pay out-of-pocket given their poor financial condition. Most caregivers did not mention the amount that they would like to pay for the TM program; only two caregivers mentioned that they would be able to pay on average PKR 3000–5000 for purchasing the proposed TM system. However, almost all caregivers mentioned that the cost of the TM program should be very minimal to maximize its reach and effectiveness.


*My brother-in-law would not be able to afford this machine due to financial constraints. I do not think he would be able to contribute to the purchase of this machine. Only yesterday, he was talking about financial difficulties and their impact on daily living. (Sister 12)*


## 4. Discussion

### 4.1. Principal Findings

This study provides an in-depth investigation into the caregivers’ perspectives and perceived acceptability of a mobile phone-based TM program to support pregnant women at high-risk for PE in Karachi, Pakistan. The study did not find differences in the challenges and perceived benefits of TM between the different kinships. The interviewed caregivers revealed several challenges in caring for high-risk pregnant women, such as frequent visits to hospitals, long travel distances, long wait hours at the clinic, disruption in the daily routine of caregivers, and high costs associated with transportation and daily blood pressure monitoring. The study identified several perceived benefits of the TM program for caregivers including a reduction in caregiver distress and anxiety, reduction in caregiver workload, increased convenience, and cost-effectiveness. The study highlighted specific caregivers’ expectations of the TM program to support high-risk pregnant women and improve the overall caregiving experience. The caregivers emphasized that the TM program should be easy-to-use and ensure the continuous monitoring of pregnant women and babies. The caregivers embraced the proposed TM program and voiced that pregnant women would be allowed to use such a program. While most caregivers believed that the TM program would be readily accepted by the wider community, a few caregivers felt that the TM program would receive resistance from less educated and non-tech savvy families and communities. Thus, the study identified both technological and cultural barriers for the acceptance of the TM program and perhaps overcoming socio-cultural barriers is more difficult. The caregivers identified the need to sensitize such families and communities through awareness sessions to improve the adoption of the TM program among high-risk pregnant women and their caregivers. The caregivers indicated that young, educated, and tech-savvy women would be able to learn and use the TM program while pregnant women with poor technological literacy may be able to receive support from husbands and other caregivers for the use of TM.

### 4.2. Comparison with Prior Research

Previous studies have informed the needs for TM programs from the pregnant women and health providers’ perspectives [12,13,14]. This study provides unique insights into the needs for TM for high-risk pregnant women from the caregivers’ perspective in a LMIC. Our study findings are consistent with previous studies conducted in high-income countries [12,13,15]. regarding the roles caregivers play in caring for high-risk pregnant women. For instance, Anders et al.’s study (2019) conducted in the United States found that caregivers performed a wide variety of actions, with the majority reporting that they accompanied the patient to medical appointments [15]. Consistent with our study, Van den Heuvel’s qualitative study (2020) conducted in the Netherlands reported that male partners faced several challenges in caring for high-risk pregnant women, including travelling to and from the hospital, feeling tiresome, and experiencing disruption in daily life [12]. Similar to our study, the Primer and Provider Selection Guide on telehealth (2013), a whitepaper developed by the Leading Age Center for Aging Services Technologies in the United States, highlighted that remote patient monitoring can provide peace of mind for family caregivers, and reduce the caregiver burdens and strains as well as bring efficiencies by reducing patient transportation costs [25]. Consistent with our prior needs assessment study with high-risk pregnant women [13], this study also reported that the TM program should be easy-to-use and ensure continuous monitoring of high-risk pregnant women and babies.

Our study highlighted some insights relevant to the future implementation of TM programs, specifically in LMICs such as Pakistan, including the inability of some women and caregivers to use the TM program and the poor acceptance of the TM system among less educated and non-tech savvy families and communities. Consistent with our findings, Dansharif et al.’s qualitative study (2021) conducted in Northern Nigeria reported several barriers associated with the implementation of TM programs for self-management in pregnancy, such as illiteracy, language barrier, lack of understanding of how to use specific apps or technology, and cultural beliefs [26]. Similar to our previous needs assessment study [13], this study identified the need to train pregnant women, as well as accompanying caregivers, on the use of the TM program and to sensitize the wider community on the usefulness of the TM program to address feasibility issues associated with technological literacy and technology acceptance. In contrast to a study conducted in a high-income country [27], the caregivers in our study showed a poor willingness to pay for the proposed TM program due to financial constraints, which may pose a challenge to implementing this care model. Growth in the field will likely depend on developing models for how to finance the mobile phone-based TM program to support pregnant women at high-risk for PE.

### 4.3. Recommendations

Based on our study findings, we offer the following recommendations for supporting informal caregivers of high-risk pregnant women in a LMIC such as Pakistan:A mobile phone-based telemonitoring program should be developed to support informal caregivers of high-risk pregnant women with the aim of reducing caregiver stress and workload, which is associated with the frequent traveling to healthcare appointments and visits to a clinic to take blood pressure readings.A context-specific TM program should be designed using a user-centric approach to include local language, visuals, and voice message alerts to allow less educated and non-tech savvy pregnant women and their caregivers to use the TM program.The caregivers should be trained on the use of TM programs to address issues associated with technological illiteracy.The caregivers should be sensitized to the benefits of the TM program to increase the acceptance and uptake of the TM program.The TM program should be low-cost to enable a large number of high-risk pregnant women and their caregivers to access the program because of the financial constraints that are common in LMICs.

### 4.4. Strengths and Limitations

A strength of this study was that multiple types of caregivers (husbands, in-laws, mothers, and sisters of high-risk pregnant women) were interviewed, which would help researchers in generalizing the findings across different types of caregivers. In addition, the majority of the caregivers interviewed were husbands, who are the primary caregivers in this context. Another strength was that the primary researcher (ASF) maintained a reflexive journal during all stages of the research to recognize and acknowledge biases during the research process. A limitation of the study was that caregivers who were interviewed were those who were willing to participate in the study and who managed to take pregnant women to the clinic for antenatal visits. The caregivers who do not take pregnant women to healthcare appointments may have different perspectives and perceived acceptability of a TM program. A second limitation was that the caregivers were only provided with a theoretical description of a proposed mobile phone-based TM program. Thus, the caregivers might not have understood the telemonitoring program completely and that might have influenced their perspectives and perceived acceptability of the program. Finally, the researchers were unable to carry out member checking with caregivers because it would have been exceedingly difficult to contact caregivers after the initial interview. However, at the end of each interview, the primary researcher restated and summarized the information obtained during the interview with the aid of interview notes, to ensure that the study data resonated with the caregiver’s perspectives and experiences.

## 5. Conclusions

The study concludes that informal caregivers play an important role and perform a rich set of activities to support high-risk pregnant women. The caregivers appreciated the proposed mobile phone-based TM program because they perceived many benefits, including a reduction in caregivers’ anxiety and workload, increased convenience, and cost-effectiveness. However, the caregivers cited several caveats to the future implementation of the TM program, including the inability of some women and caregivers to use the TM program and the poor acceptance of the TM system among less educated and non-tech savvy families and communities. The caregivers suggested that the TM system should be easy to use and should ensure the continuous monitoring of the pregnant women and the baby. The findings from this study will be directly used to design the context-specific mobile phone-based TM program and implement a feasibility trial to support high-risk pregnant women and improve the overall caregiving experience.

## Figures and Tables

**Figure 1 healthcare-11-00392-f001:**
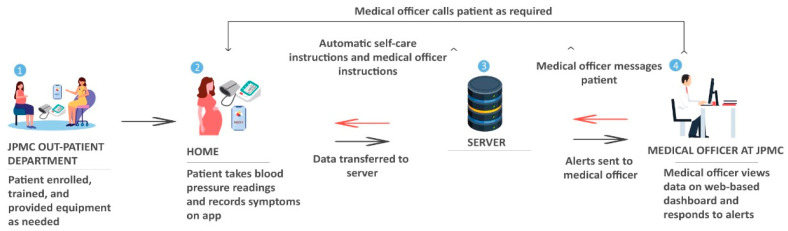
Proposed patient and clinician workflow for the telemonitoring program at JPMC.

**Table 1 healthcare-11-00392-t001:** Eligibility criteria for caregivers of pregnant women at high-risk for PE.

Inclusion Criteria	Exclusion Criteria
Caregivers of pregnant women who met the National Institute for Health and Care Excellence (NICE) guidelines [22] definition of high-risk for PE (Appendix A) Caregivers of pregnant women who were at least in their follow-up visit after being identified as high-risk for PE and thus have had time to reflect on the disease condition, and associated difficulties. Caregivers of high-risk pregnant women with the ability to speak the Urdu language.	Caregivers of high-risk pregnant women who were admitted to the in-patient wards and emergency care for treatment purposes

**Table 2 healthcare-11-00392-t002:** Characteristics of caregivers (n = 28).

Characteristics of Caregivers	Category	N (%) or Mean ± SD
Gender	Female	13 (46.42%)
	Male	15 (53.57%)
Age		34.82 ±11.48
Educational Level	No education	9 (32.14%)
	Less than high school	9 (32.14%)
	High school	8 (28.57%)
	College or university	2 (7.14%)
Occupation	Homemaker	12 (42.85%)
	Professional	16 (57.14%)
Relationship with pregnant women at high-risk for PE	Husband	15 (53.57%)
	Mother	2 (7.14%)
	Mother-in-law	2 (7.14%)
	Sister	6 (21.42)
	Sister-in-law	1 (3.57%)
	Wife of Husband’s brother	2 (7.14%)
Caregiving experience (years)		2.42 ± 1.19
Access to a mobile phone	Yes	25 (89.28%)
	No	3 (10.71%)
Type of Phone	Basic mobile phone	12 (42.85%)
	Smartphone	13 (46.42%)
Access to the internet	Yes	10 (35.71%)
	No	18 (64.28%)
Availability of blood pressure monitor at home	Yes	19 (67.85%)
	No	9 (32.14%)

**Table 3 healthcare-11-00392-t003:** Themes and categories.

Themes	Sub-Themes
Caring for pregnant women at high-risk for PE	Caregivers’ roles in supporting high-risk pregnant women
	Challenges in caring for pregnant women at high-risk for PE
Supporting caregivers through the use of the TM program	Reducing caregiver distress and anxiety
	Reducing caregiver workload
	Convenient and cost-effective
Caregivers’ expectation of the TM program	Ease-of-use of TM program
	Continuous monitoring of pregnant women
Considerations for implementing the TM program	Caregivers and community acceptance of TM program
	Training of pregnant women and caregivers on TM program use
	Cost of TM Program

## Data Availability

The datasets used and/or analyzed during the current study are available from the corresponding author on reasonable request.

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
