# Peer review of "Exploring Caregivers’ Perspectives and Perceived Acceptability of a Mobile-Based Telemonitoring Program to Support Pregnant Women at High-Risk for Preeclampsia in Karachi, Pakistan: A Qualitative Descriptive Study"

_healthcare, 2023, doi:10.3390/healthcare11030392_

Round 1
Reviewer 1 Report
The present study provides an investigation into the caregivers' perspsectives and acceptability of a mobile phone-based tele monitoring program to support pregnant women at high-risk for preeclampsia in Karachi, Pakistan. The surveyed caregivers revealed several challenges in caring for high-risk pregnant women such as frequent visits to hospitals, long travel distances, wait hours at the clinic, disruption in the daily routine of caregivers, high costs associated with transportation...
Then, the article shows an interesting application of mobile technology that could presumably be both convenience and cost-effective. In fact, the tele monitoring program for caregivers include a reduction in caregiver distress and anxiety, as well as a reduction in caregiver workload. Nevertheless, the paper should further develop the advantages of this technology for pregnant women. They can be indirectly deducted, but need to be made explicit. More data about preeclampsia morbidity should also be appreciated.
The paper mentions two types of barriers: technological and cultural. Some barriers are associated with the implementation of tele monitoring programs for self-management in pregnancy such as illiteracy, language barriers or lack of understanding of how to use specific apps or technology. Perhaps more difficult to overcome are socio-cultural barriers.
Finally, there are some mistakes in expressions or quoting references. For instance:
. was used 13 to explore caregivers' perspectives
. sampling technique 15 16 to recruit caregivers
. study 18, our study used digital audio
. approach 19 was used
. Wife of husband's bother
. providers' perspectives 7-9
. high-income countries 7 8 10 regarding
. medical appointments10
. in daily life8
. transportation costs20
. pregnant women7
. cultural beliefs21
. assessment study 7
. high-income country22
Author Response
Thanks so much for appreciating the work and highlighting the mistakes in the references. Please note that the corrections have been made in all the references. Data on preeclampsia morbidity is limited in Pakistan and thus we are unable to provide more specific morbidity-related data. We have now explicitly mentioned in the discussion section that 'the study identified both technological and cultural barriers for acceptance of TM program and perhaps overcoming socio-cultural barriers is more difficult (line no 422-423). Please let us know if any further changes are required.
Reviewer 2 Report
Thank you for your work in this very important topic. I really enjoyed reading your work.
The paper is well designed and discussed in interesting way. The introduction provides sufficient background and include all relevant references. The research design is appropriate. The methods are adequately described. The conclusions are supported by the results. The authors' discussion of the research is conducted in an interesting and professional way. The authors have identified valuable recommendations.
I accept the article after minor revision:
1) The Authors should add the DOI in the reference list.
2) The authors should check and correct the text, e.g. in line 94 is"13" (not [13]). Is it a quote?
I propose to accept this scientific article for publication due to its high scientific value. The manuscript might be of interest to the journal's readership.
Author Response
Thanks so much for appreciating the work. Please note that corrections have been made in the references. As instructed by the journal, we have used Endnote software and numbered reference format to do the referencing of the paper which does not allow us to enter the DOI number. Please let us know if any further changes are required. Thank you for your time.
Reviewer 3 Report
The article “Exploring Caregivers’ Perspectives and Perceived Acceptability of a Mobile-Based Telemonitoring Program to Support Pregnant Women at High-Risk for Preeclampsia in Karachi, Pakistan: A Qualitative Descriptive Study” addresses a very relevant topic, contributing to a better understanding on how caregivers perceive telemonitoring programmes to support pregnant women.
The Introduction section presents the main theoretical concepts that underpinned this article, and the references are up-to-date and coherent with the article topic. However, there are some arguments that would be important to develop in order to improve the quality of the manuscript:
- Authors refer that “The needs assessment study also revealed several socio-cultural barriers that could potentially influence the use of the proposed TM program” (p. 2). What socio-cultural barriers?
- In page 2, it is argued that “Given the prominent role that caregivers play in the lives of pregnant women, we explored caregivers’ perspectives and perceived acceptability of a mobile phone-based TM program”. I believe one of the strongest aspects of this manuscript is considering caregivers’ perspectives, however, it would be interesting to have a clear definition of who can be considered a caregiver, in this case, an informal caregiver, the role they assume, and its consequences. This would help justify the importance of telemonitoring devices in the caregiving process.
Regarding the materials and methods section, authors made very clear the several steps that were taken during the study. The research design appears to be in accordance with the aims of the article. Yet, it could be made clearer if each interviewed caregiver correspond to one pregnant woman, or if there were caregivers taking care of the same woman.
The results section is organized considering the main topics that emerged from the analysis and is essentially descriptive. For each theme there are quotes from the participants, which enriches the section. However, in some cases it would be interest to have more examples and from different type of caregivers. For instance, theme 3.1.1 only presents quotes from one husband.
Since the results section is very descriptive, I suggest the discussion section to be further developed, for instance, do the authors found differences between the different kinships? The recommendations appear again in the conclusions section, so maybe authors can avoid the repetition. I appreciated the Strengths and Limitations section. Authors mentioned that “A second limitation was that the caregivers were only provided with a theoretical description of a proposed mobile phone-based TM program. Thus, the caregivers might not have understood the telemonitoring program completely and that might have influenced their perspectives and perceived acceptability of the program” (p.12). I agree with this, and also, I think it would also benefit the manuscript if the theoretical description of the proposed mobile phone-based TM program caregivers had access was available. Finally, the role and availability of healthcare professionals could be problematized.
In conclusion, the article is clear, relevant for the field and well-structured. I recommend the publication of this article after minor revision.
Author Response
Thanks so much for appreciating our work. We agree with most of the arguments/comments and have made some track changes throughout the paper to address these comments. Please see the point-by-point response below:
1. Comments related to the introduction section
Response: To address the first comment in the introduction section related to sociocultural barriers, the following additions have been made in track mode in line no. 77-85
"The needs assessment study also revealed several socio-cultural barriers that could potentially influence the use of the proposed TM program including gendered imbalances in decision-making, restricted access to education and healthcare, and restricted women's mobility. The interviewed high-risk pregnant women, nurses, clinicians, and digital health experts in the needs assessment study reported that pregnant women face restrictions, from in-laws and husbands, regarding the use of mobile phones and the internet. In addition, the study highlighted that women would require permission from husbands and in-laws before opting into the TM program, and family members might discourage the use of a TM program to support high-risk pregnant women (14).
To address the second comment in the introduction section related to informal caregivers, the following additions have been made in track mode in line no. 88-91.
"In this study, informal caregivers refer to family members including male partners, mother-in-law, sister-in-law, and others in the family who provide unpaid assistance related to pregnancy care and support such as transportation and personal care."2. Comments related to the methods and material section
Response: To address the comment in the methods section related to caregiver interviews, the following additions have been made in track mode in line no. 158-160
"We anticipated conducting and recording 20-25 interviews with informal caregivers of high-risk pregnant women; however, our study conducted 28 interviews with caregivers in order to achieve data saturation. Each interviewed caregiver corresponds to one pregnant woman."
3. Comments related to the results section
Response: Thank you for the comment related to the quotes in the results section. Based on journal requirements, we have ensured that each sub-theme has at least 2 quotations. In cases, where there are fewer quotations... it is due to data limitations.
4. Comments related to the discussion section
Response: Thank you for providing suggestions in the discussion section. We have now noted that the study did not find differences between the different kinships (line no. 415-416). We have removed the limitations from the conclusion section. We have provided the theoretical description of the proposed mobile phone-based TM program in the method section (113-127). Given, the scope of this paper was informal caregivers, role and availability of healthcare professionals is not mentioned. However, the perspectives of providers have been explored separately and published here https://pubmed.ncbi.nlm.nih.gov/35200152/
Please let us know if any further changes are required in the paper.
Thank you,
Anam